# Using a Mobile Application for Health Communication to Facilitate a Sense of Coherence: Experiences of Older Persons with Cognitive Impairment

**DOI:** 10.3390/ijerph182111332

**Published:** 2021-10-28

**Authors:** Erik Piculell, Lisa Skär, Johan Sanmartin Berglund, Peter Anderberg, Doris Bohman

**Affiliations:** 1Department of Health, Faculty of Engineering, Blekinge Institute of Technology, 371 79 Karlskrona, Sweden; lisa.skar@bth.se (L.S.); johan.sanmartin.berglund@bth.se (J.S.B.); peter.anderberg@bth.se (P.A.); doris.bohman@bth.se (D.B.); 2Department of Health Sciences, Skövde University, 541 28 Skövde, Sweden; 3Optentia Research Unit, North-West University, Potchefstroom 2531, South Africa

**Keywords:** cognitive impairment, health communication, mobile application, older persons, sense of coherence

## Abstract

The increasing use of technology by older persons and their preferences for living at home and being independent have created an avenue for self-care and care delivery using mobile technologies and health communication. This study aimed to explain how older persons with cognitive impairment experienced technology-based health communication through the use of a mobile application to facilitate a sense of coherence. Individual, semi-structured interviews with 16 participants in the SMART4MD project were conducted. The interviews were transcribed then coded deductively and thematically, creating themes that corresponded to the central components of the sense of coherence model: comprehensibility, manageability, and meaningfulness. The findings produced an overall theme: a challenging technology that can provide support, based on the three identified themes: making sense of mobile technologies, mastering mobile technologies, and the potential added value to use mobile technologies. The participants’ experiences were influenced by their previous use and expectations for the application. Personal support, cognitive and physical ability, and different sources for information impacted use. The participants experienced that using the application created an ambiguity to be challenging and have possible benefits. The study suggests that the sense of coherence model may be used as a method to understand the use of technology by older populations.

## 1. Introduction

As the use of technology can facilitate an independent life in old age, users of technology need to possess adequate knowledge and skills (i.e., technological literacy (TL)) [1], encompassing the ability to interact with and understand technical products [2]. Older persons are increasingly using mobile technologies as communication tools, such as technology-based health communication (HC), which has been shown to assist in finding information [3,4,5,6]. Research on mobile assistive technologies used by older persons with cognitive impairment (CI) has found that tablets and smartphones may benefit such users but should be adapted to their needs [7]. In this study, the term “older person” is defined as 55 years old and above following the inclusion criteria of the Support Monitoring and Reminder Technology for Mild Dementia (SMART4MD) project, from which the participants in the current study were recruited. Mobile applications may assist and aid older persons who have functional and CI, including difficulties with attention, working memory, and the ability to learn new information [8,9,10]. Therefore, older persons’ use and perceptions of technologies are influenced by personal, physical, and social factors [11]. Due to aging, physical and CI can make the use of technologies difficult [8], highlighting the importance of individual preferences and needs in development of new technologies [12]. Research on older persons and HC has shown that older persons prefer health information to be credible and easily understood. Therefore, to develop usable technology and account for possible CI, the design of technology-based HC should be grounded in user experiences [13,14,15].

The sense of coherence (SOC) model helps understand life satisfaction in old age [16]. The SOC, developed by Antonovsky [17,18], builds on three central components: comprehensibility, manageability, and meaningfulness. The model reflects how a person perceives the world and uses their own and external resources to manage different challenges (or stressors), which either hinder or contribute to a sense of coherence and health. Comprehensibility reflects how comprehensible challenges or stressors are seen. Manageability reflects how manageable these challenges are considered. Meaningfulness indicates how meaningful these challenges are viewed. SOC has previously been applied as a framework for analysis to understand older persons’ use of mobile applications from a quantitative approach [19]. However, qualitative research that targets the use of mobile technologies by older persons with CI and that employs SOC as a theoretical model remains scarce [20]. Additional qualitative studies using the SOC model as an analytical framework are needed [21]. SOC was, therefore, chosen in this study as a suitable model to enhance the understanding of older persons’ technology use to improve their health.

Research has shown that while older persons increasingly use and have access to technology, more suitable, user-friendly solutions are necessary to address the needs of individuals with CI [15]. As the shift toward home-based health care is supported by technological solutions and HC, designs based on older persons’ actual needs and experiences are required [22]. Hence, the aim of this study was to explain how older persons with CI experienced technology-based HC through the use of a mobile application to facilitate a SOC.

## 2. Materials and Methods

### 2.1. Design

This qualitative study adopted a deductive approach to explain how older individuals with CI experienced technology-based HC using a mobile application to facilitate a SOC. Data were collected using semi-structured interviews, analyzed and sorted into themes using a matrix [23]. The themes corresponded to the description of each central components in the theoretical SOC model: comprehensibility, manageability, and meaningfulness. A deductive approach was chosen because the study aimed to understand older persons’ experiences better using the theoretical model.

### 2.2. Study Context

The EU project SMART4MD is a multicenter project aimed at investigating the effects of an intervention consisting of a mobile application designed for and used by older persons with CI, described further in Anderberg et al. [24]. In this study, the term “mobile application” refers to the software and hardware specifically used and developed for the SMART4MD project. The software application was installed on a provided tablet with a seven-inch screen and had different functionalities, such as the possibility of setting medication reminders, obtaining information about dementia, reading web-based newspapers, and playing games (see Appendix A). Participants in the SMART4MD project scored 20–28 on the Mini-Mental State Examination (MMSE), were 55 years or older, had experienced memory problems in the six months prior to the project, did not receive any formal care, had no functional disability affecting the use of the mobile application, administered their medication, were not depressed, and had an informal caregiver who participated in the project.

### 2.3. Participants

In the current study, participants from the Swedish site of the SMART4MD project were recruited purposefully based on the following inclusion criteria: an MMSE score ranging from 20–26, an indication of CI, and access to the mobile application for more than one year prior to the study. The participants had also acknowledged problems with memory recall. The participants were contacted by phone by the first author (EP), informed verbally about the project, and given a verbal description of the aim of the interviews. After the call, information letters were sent out in advance either by regular mail or by e-mail, according to the participant’s preference. When an individual agreed to participate, a time for the interview was scheduled. In total, 16 participants were recruited, comprising four females and 12 males aged 71–100 (see Appendix B for participant characteristics).

### 2.4. Data Collection

Individual, semi-structured interviews were conducted in June–August 2019. Each interview lasted between 17 and 55 min (mean = 34 min). An interview guide (Appendix C) was devised to ensure that the same topics were discussed with all participants and that there would be opportunities to probe specific subjects further [25]. The interview guide was partially based on Robinson et al. [26]. The interviews commenced with questions regarding the participants’ previous knowledge of technology use. These were followed by questions about their mobile application use with follow-up prompts, such as “Could you give me an example?” or “Can you elaborate further?” The first interview was conducted as a pilot interview to ensure the clarity, adequacy, and appropriateness of the interview questions [27,28]. The pilot interview was included in the data, with no changes to the interview guide. All the interviews took place in the participants’ homes and were conducted, recorded, and transcribed verbatim by the first author (EP), a PhD student with previous experience of conducting interviews at the bachelor’s and master’s degree levels.

### 2.5. Data Analysis

The transcribed interviews were read to gain a general sense of the whole and units of analysis responding to the aim of the study were marked. The analysis was based on the SOC to understand the participant’s use of mobile technologies. To analyze the experiences, the units of analysis were grouped using a matrix as described by Elo and Kyngäs [23]. The matrix was employed to sort the data corresponding to each central component in the theoretical SOC model: comprehensibility, manageability, and meaningfulness [17,18]. These three areas were subsequently analyzed thematically, following the six phases described by Braun and Clarke [29]. The six phases were as follows: (i) familiarization with the transcribed interviews; (ii) generation of initial codes from the units of analysis; (iii) creation of subthemes of the codes and the relationships among them; (iv) review of the themes corresponding to the description of the three components of the SOC theoretical model; (v) defining and naming the themes; and finally (vi) production of the report. Phases one and two were conducted in parallel with the deductive coding. The analysis was conducted by moving back and forth between the phases as needed. All the authors carried it out to obtain consensus in sorting the data into related themes according to the descriptions of the SOC model components. The analysis ended with the creation of a thematic map explaining the interrelations between the overall theme and the different themes corresponding to the SOC model.

### 2.6. Ethical Considerations

This study was conducted in accordance with the Declaration of Helsinki [30]. Each regional ethical review board granted ethical approvals for the SMART4MD project at each participating site to comply with research regulations in the respective countries. Approval for the Swedish site was granted by the Regional Ethical Review Board, Lund (code BTH:SMART4MD dnr 2016/470), and included approval for subsequent studies within the project. The written consent of the participants in the SMART4MD project was collected during their enrollment. As part of the informed consent process for the project, written information in the form of a participant information sheet was sent out at least 24 h prior to a screening meeting. Verbal and written consent were obtained during this screening meeting with the research team and the participant’s informal caregiver present. The research team that assessed the participant’s capacity to consent to the SMART4MD project of the Swedish site comprised of personnel with education in caring science, research nurses, and PhD students. The consent process was renewed during the project in follow-up visits every six months. More detailed information on the screening and consent process in the SMART4MD project is available in Anderberg et al. [24].

The participants’ capacity to consent in the present study was determined by asking the participants for and receiving their consent multiple times and the consent already given for the SMART4MD project. The participants were provided with information about the purpose of the interviews, both verbally and in writing, by mail or e-mail. Before commencing of the interviews, this information was repeated if necessary. The participants were also asked if they had any further questions and were informed of their right to withdraw from the interview at any time [28]. Participants also gave verbal consent and permission to record the interviews before the interviews started. Striving for the best conditions and considering the participants’ CI, they needed to understand the provided information to give their consent. The participants’ informal caregivers were also present to offer assistance if needed, but they were not in the interview room. The first author, EP, who assessed the capacity to consent, has previous experience with the consent process within the SMART4MD project, in thesis work at bachelor’s and master’s degree levels, and from a course in good clinical practice. The interview settings were intended to ensure a familiar and secure environment for the participants, and the interviews were conducted in their homes. The recorded interviews and details about the participants were only accessed by the first author (EP) and kept at a safe location to avoid unauthorized access in accordance with the General Data Protection Regulation (GDPR) [31] and Personal Data Act [32].

## 3. Results

The analysis based on the components of the theoretical SOC model resulted in the three following themes: making sense of mobile technologies; mastering mobile technologies, and the potential added value to use mobile technologies. In addition, the analysis also gave rise to the following overall theme: a challenging technology that can provide support, which corresponds to ambiguity in the SOC model. The thematic map (see Figure 1) depicts the overall theme and the interrelation between the different themes and the overall theme, illustrated by arrows. The themes are described in the sections below using quotations from the interviews. Each quotation has been anonymized to maintain confidentiality.

### 3.1. Themes

#### 3.1.1. A Challenging Technology That Can Provide Support (Sense of Coherence)

The overall theme explains the participants’ views of the mobile application as being both challenging and potentially supportive, depending on their needs and interests. The challenges caused hesitation and doubtfulness due to the participants’ frustrations arising from the application’s inability to function as expected. The overall theme encompasses the three themes, highlighting that HC needs to be understandable, user-friendly, and coherent through mobile applications.

#### 3.1.2. Making Sense of Mobile Technologies (Comprehensibility)

This theme describes the participants’ views of the mobile application as understandable and comprehensible as well as challenging. The theme contains the following two subthemes: understanding usage and importance of language proficiency.

##### Understanding Usage

Participants expressed that it could be difficult to understand the mobile application, use its different functionalities, and access its information. The participants were open to other technologies, such as smartphones, which they frequently used. The text in the application, such as about dementia, was regarded as clear and informative. Sometimes, however, the participants had difficulties accessing the content.


*“I may not have found all the features. It’s very, very likely”.*
(Participant A)

The participants also noted the evolution of mobile technologies, and the need to relearn these technologies made them feel left out. Although the participants had previous technical knowledge and work experience with computers, they felt that they could not keep up with software and features that changed over time.


*“So, I can’t access and do as I did before”.*
(Participant F)

The participants also feared that misuse of the mobile application could lead to consequences due to a lack of understanding of the application or incorrect use. Incorrect use could result in errors and, in turn, a lack of trust in technology.


*“Uncertainty makes me afraid to make mistakes, which will have consequences”.*
(Participant N)

##### Importance of Language Proficiency

Language, especially English, was considered important when using the mobile application. The application itself was in Swedish. However, information not in the participants’ native language, such as notifications in English in the tablet’s operation system, could lead to uncertainty. When the participants did not understand the English vocabulary used in the operating system, they needed to ask for help from, for example, a spouse, next of kin, or close friend who could help and translate for them.


*“I am not 100 in English, but then I have help from upstairs. She knows that if there are certain words or a sentence, then I can ask her”.*
(Participant G)

Another example of the uncertainty and challenges surrounding language was when Swedish was not the participant’s native language. Even though the participant could read Swedish, they experienced challenges using the correct spelling and avoiding being misunderstood.


*“I must write here that I have heart pain for example … how can I do that … for example, I have symptom heart failure … how can I do that … enter here … I cannot … must write heart … I can spell so wrong … I am not Swedish”.*
(Participant M)

#### 3.1.3. Mastering Mobile Technologies (Manageability)

This theme describes the participants’ views on having the resources available to utilize mobile technologies. The theme contained the following three subthemes: ability to use the interface, expectations on functionalities, and need for support.

##### Ability to Use the Interface

Reading the display of the mobile application and using the touchscreen to touch and scroll were considered difficult. The participants also found the application complicated to learn, adding that worsening memory problems could make the use of technology even more challenging.


*“Partly I have some difficulties seeing too. That’s not so good for me … And so, it’s this with the touch buttons and this”.*
(Participant P)

However, the participants also noted that even if they made mistakes when using the application, they could do it repeatedly because learning how to use technology requires patience.


*“Sometime you do … I am making mistakes, so something else comes up but… then you have to go back and do it over … again because … everything is not perfect at once”.*
(Participant E)

##### Expectations about Functionalities

The participants expressed facing difficulties and frustration when using the application due to software restrictions and slow devices. When the tablet was slow or did not work as anticipated, other kinds of technologies, such as mobile phones or computers, were preferred.


*“It’s the one I used and the reason I did not continue … I think was that I thought this [the mobile application] was quite slow”.*
(Participant A)

##### Need for Support

The participants shared the view that they preferred to have supervision from someone, such as a family member or a close friend, who understood how to use the mobile application. Such help could be in the form of guidance while using the application or on which button to press. If the participants had trouble finding someone to explain or help, they tended to view the application as challenging to operate. They also highlighted the comfort of having someone who could provide support for the application and other technologies. According to the participants, it was easier to ask for help from, for example, a spouse or other family member and resolve the problem faster than to try to decode the application by themselves.


*“If something shows up then well … then I have to contact [my spouse] about which button to use”.*
(Participant E)

#### 3.1.4. The Potential Added Value to Use Mobile Technologies (Meaningfulness)

This theme describes the participants’ views on the ability of the mobile application to satisfy their needs. The theme contained the following three subthemes: the importance of fulfilling needs, choice of information sources, and importance of personal interaction.

##### The Importance of Fulfilling Needs

The participants observed that the different functionalities of the application could be useful. For example, according to the participants, the use of other technologies could be an option to access health care, but they currently used their telephone, which lead to waiting in telephone queues. The application was further seen as a helpful way to access, share, and receive information and set reminders.


*“It’s very good that you can get … notifications when you need them”.*
(Participant L)

However, if the participants did not perceive that the application fulfilled a need or considered it unbeneficial, their interest in using the application decreased further. One participant expressed disinterest in the reminder feature due to the lack of a sense of need.


*“Yes, I have accessed and viewed most of it, but then I haven’t … I haven’t been that interested”.*
(Participant P)

The lack of interest in using technology was also ascribed to participants viewing themselves as old or old-fashioned. The participants stated that there are probably other older persons interested in using technology, just not just themselves.


*“I am probably a bit old-fashioned to use that. Difficult to learn this with the touch function”.*
(Participant I)

##### Choice of Information Sources

The participants preferred other ways of finding information and setting reminders, such as methods for documenting or searching for information that were considered more convenient. Taking notes with a pen and paper and using notebooks and physical calendars were preferred to the mobile application. Participants also used paper to list their daily tasks and check off items that had been accomplished during the day. Other familiar technologies, such as other tablet models or desktop computers, were considered easier to use and hence more useful than the application. Participants mentioned preferring familiar technologies, such as other kinds of devices or software, for which they did not have to relearn to perform similar tasks.


*“No, I’m using the computer … because it’s easier than the tablet … it’s easier, quicker, not the same system”.*
(Participant B)

However, the mobile application was considered suitable for different tasks, such as for medication reminders. The bigger screen on the tablet was also seen as positive. Yet, the participants also had smartphones with them with the functionalities they needed, and these smartphones were considered more convenient due to their smaller size and ability to fit in a pocket. Moreover, a smartphone can be used to make phone calls and is more portable than a tablet, which was considered more suitable for use within the home. The participants further mentioned that they did not use the reminders in the application when they were not at home because they did not take the tablet out with them.


*“No, I think it is too big to have in the bag … then you would need a bigger bag … and I do not have my medications with me”.*
(Participant L)

These limitations caused uncertainty regarding the application’s purpose, especially as smartphones were considered more practical despite their smaller display size. Moreover, owning different mobile technologies that served similar purposes resulted in having too many devices to carry around.


*“It has worked well, but it’s just that you think you have it twice there, so it is a little easier to use”.*
(Participant O)

##### Importance of Personal Interaction

The participants emphasized the importance of meeting a health care professional in person rather than using technology. The participants felt safer and taken more seriously when sharing their concerns with a professional in person. They also had a greater understanding of the information given to them by health care professionals in person rather than during video calls.

*“To speak with a doctor and look them in the eyes instead of speaking on the phone … I think it’s important to meet them”.* (Participant E)

## 4. Discussion

This study aimed to explain how older persons with CI experienced technology-based HC through the use of a mobile application to facilitate a SOC. The findings show that the use of the mobile application created an ambiguity, as it was both challenging and had possible benefits. These findings are summarized in the overall theme of “a challenging technology that can provide support”. This finding aligns with Pirhonen et al. [33], who observed that older persons considered technology to have both advantages and drawbacks. Furthermore, Hedman et al. [7] found that technology is viewed as complex and multifaceted and greatly impacts the daily activities of older persons with CI. In the study, participants expressed that for the application to be valuable, it had to both be user-friendly and fulfill a need. When participants considered that learning to use the application was not worth the effort, they showed low interest in further engagement. This can be interpreted as that the mobile application did not have the desired or expected benefits. Previous research has uncovered barriers that older persons face regarding technology use, such as high cost, lack of interest, lack of guidance, and device complexity [34]. In the SOC model, all three central components should be viewed positively [18]. The three themes of the study’s findings are discussed in relation to the SOC model components: comprehensibility, manageability, and meaningfulness, as illustrated in the thematic map. The interaction between the themes and overall theme in the findings corresponds to the relationship between the SOC model and its central components.

The theme “making sense of mobile technologies” reflected the comprehensibility of the mobile application. The first subtheme, “understanding usage”, focused on the participants’ views of the application as difficult to use to find information and their concerns about the evolution of technology and fear of its consequences. Hedman et al. [7] supports these finding that updating software can be perceived as difficult and impossible to avoid. Furthermore, older persons’ own beliefs about being incapable of learning contributed to their fear of using technology and making mistakes [7,35]. However, older persons could be confident in using technology in some situations, implying that the difficulties in technology use are related to impairments and not age itself. The second subtheme, “importance of language proficiency”, focused on the language when using the mobile application, as the operating system was in English. Language proficiency is an aspect of functional literacy, which comprises reading and writing [36]. Functional literacy serves as the basis for other forms of literacy, such as TL. In the findings, the use of the application constituted a dual complexity related to language and the CI. Implying that language as a cultural difference is related to health technologies, Matthew-Maich et al. [37] emphasized considering cultural differences and values when developing technological solutions. However, older persons with cognitive and visual impairments are less likely to use technology [8].

A sense of comprehensibility is achieved in the SOC model when both inner and outer stimuli are considered coherent, clear, and structured [18]. Thus, comprehensibility in the SOC model is the second most crucial component after meaningfulness because understanding affects comprehensibility. This relation is also viewed in the findings, in which the comprehensibility of the mobile application impacted the participants’ interest or disinterest in further use.

The theme “mastering mobile technologies” reflected the ease of use of the mobile application and its functionalities. This was influenced by the participants’ previous knowledge and personal skills in using technology. The first subtheme, “ability to use the interface”, focused on the difficulties encountered while using the touchscreen, patience of using new technologies, and memory loss. These findings are consistent with those of Bogza et al. [38], who observed that difficulties encountered while navigating technologies resulted in frustration in decision making. Hence, the information presented to persons with CI has to be meaningful, concise, and easy to remember. In the second subtheme, “expectations about functionalities”, the slowness of the application also contributed to frustration when the application’s functionality was not an expected benefit. Even with a simplified interface easy to use, the application’s responsiveness affected perceptions, and the device’s slowness was found to be frustrating.

Previous research support that low engagement using mobile technology was due to difficulties of responsiveness of touch-screen devices [39]. Regarding the third subtheme, “need for support”, personal support was seen as valuable when using the application. This finding is also confirmed in previous research, including informal caregivers’ support [40,41,42]. Pirhonen et al. [33] argued that availability of resources rather than abilities of older persons can explain access to and use of digital technologies. Blok et al. [43] explained that a low interest in the use of technology is related to difficulties in asking for help. The participants’ attitudes toward technology depended on their quality of life experience and sociodemographic variables, presented in another study [44]. Socioeconomic factors play an important role regarding the “digital divide”, as the use of smartphones and access to other technologies is necessary for digital inclusion [45]. The findings highlight the challenges imposed by technology as a means of HC. The fact that smartphones are not available to everyone has to be considered when implementing mobile applications and digital channels for older persons [46]. A recent study in which tablets were used for the cognitive training of older persons with CI showed promising results for HC mediated by mobile technology on cognitive ability but not on depression and daily activities [47]. Other studies, including technology-based cognitive training, have pointed to a positive attitude in older persons regarding technology and suggested adaptions of a simplified user interface and instructions and the inclusion of reminders [39,48]. Education may contribute to the use of technologies of persons with a lower level of education, which correlates with a lower use of health technologies [40]. Older persons prefer health information that is credible [14]. Therefore, it must be acknowledged that the participants in this study had prior experiences using different kinds of technology, such as computers, smartphones, and tablets.

The theme “the potential added value to use mobile technologies” reflected participants’ views of the mobile application as meaningful and with different preferences for information sources. Regarding the first subtheme, “the importance of fulfilling needs”, the application had to meet a need to be considered interesting to use, which is consistent with previous findings in which technology was observed to contribute to self-management [12]. Control, interactivity, and perceived usefulness have been pointed out to influence older persons’ use of technology [49]. Additional factors include security, independence, safety, and the ability to socialize and receive support in daily activities apart from health management [50]. Mercer et al. [51] further pointed out the significance of understanding what motivates technology adoption. A sense of meaningfulness is significant for motivation for older persons and is central in the SOC model [18]. The second subtheme, “choice of information sources”, focused on participants’ use of different information sources and means of communication involving both mobile technologies and paper-based information. Using traditional media, such as notes and magazines, has been previously confirmed as beneficial in complementing web-based information [52]. In the findings, smartphones were preferred to tablets due to their smaller size, which made them more portable and had perceived additional functionalities. However, previous research has indicated that persons with CI who use mobile technologies found big screens more suitable, especially with the possibility to enlarge the graphics [53]. Due to enhanced portability, it may result in preferences for smaller mobile devices [54]. The third subtheme, “importance of personal interaction”, highlighted how social interaction was valued, which is consistent with previous research on technology use by older persons [55]. These findings are also supported by Borg et al. [45], who emphasized the need for social interaction with family and friends and supporting skills to ensure digital inclusion. Using technology contributes to social contact and improved involvement in self-care of older persons [56,57,58]. Obtaining support from formal and informal caregivers can improve the utilization of HC in a home environment [59]. Ethical considerations when implementing technology for older persons also have to be highlighted [55], mainly due to the vulnerability related to CI. The present study has emphasized considerations in developing a communication path that contributes to independence in home-based care. The findings advance knowledge about how older persons with CI use HC to increase their independence regarding home-based care.

The relevance of meaningful technologies to independence is considered necessary in future home care [60]. Bol et al. [61] indicated room for improvement regarding tailored technology-based HC. Social support and collaborative design (co-design) have been identified as strategies to improve digital inclusion and reduce barriers to technology use, such as attitude, digital ability, and access [45]. Furthermore, both the design and functionalities of technologies should fit older persons’ cognitive and physical profiles, acknowledging their variety of needs and requirements [62]. As CI impacts individuals’ ability to seek information, technology for the user needs to include specific functionalities, such as avoiding scrolling and the use of horizontal browsing [63]. Research has underscored specific usability aspects related to the use of mobile technology, such as limitations related to the size and design [64,65]. However, to avoid challenges related to the use of technology-based HC, it is imperative to consider individual skills, preferences, and characteristics and the context in which the applications are used when developing mobile applications [54]. Previous research supported this, emphasizing that social support benefits HC, such as information seeking [66]. Thus, the presence of social support and level of CI affects the use of technology-based HC.

When applying the SOC model to the use of technology by older persons, the meaning of TL can be interpreted as similar to comprehensibility and manageability in the SOC model (i.e., the ability to use and understand technology). When utilizing HC, the concept of health literacy (HL), which signifies the understanding, use, and perception of health-related information [36], also impacts the interpretation of the communicated content. In the findings, both TL and HL were important skills when using mobile technologies. Technology-based HC can contribute to older persons’ satisfaction of emotional and social needs [43]. Because technology can improve older persons’ self-care and access to HC, research has confirmed the importance of TL and personal support in using technologies [67]. HL is further necessary to protect against the risks of misinformation when using digital channels for HC [68]. The findings in this study suggest that the SOC model may contribute to a deeper understanding of technology-based HC among older persons with CI.

### Methodological Considerations

The findings of this study offer important information regarding the experiences of older persons with CI. Regarding trustworthiness [69], the authors continuously discussed how to enhance the analysis credibility (and confirmability). All authors were involved throughout the analytical process, and the themes were discussed until a consensus was obtained. To improve credibility and transferability, the analysis process has been described in detail and illustrated with quotes. In addition, an interview guide was used to increase dependability, which is preferable in semi-structured interviews [25].

However, this study also has limitations. Despite continuous probing and descriptive questions during the interviews, highly detailed responses were scarce. For example, the participants expressed views about the mobile application and tablet without giving further explanations. The study’s deductive approach may also have produced a less detailed description of the data and affected credibility. Due to CI, the participants may have had difficulties expressing their views on using the mobile application and their perceptions of HC, which may affect dependability.

Furthermore, other age-related challenges may have contributed to the participants’ experiences of using the mobile application, although functional impairment was an exclusion criterion in the SMART4MD project. A feasibility study was conducted to improve the application in the SMART4MD project [70], and the final version was used in this study. Lastly, because the participants’ CI in the SMART4MD project was based on MMSE scores, the scores might not have been an accurate indication of the participants’ cognitive ability at the time of the interviews for the present study.

## 5. Conclusions

This study indicated that using a mobile application for technology-based HC created an ambiguity to be challenging and have possible benefits. This was challenging for older persons with CI and affected their engagement. Mobile technology was related to the perception of being helpful, easy to use, and fulfilling of needs. The participant’s differences in abilities affected preferences, relevance, and choice of HC sources, either mediated by technology or not. The participant’s skills and expectations contributed to perceived benefits. Having support with the application contributed to feelings of meaningfulness and interest and thus motivation. Personal support also improved the usefulness of technology-based HC. However, personal interaction with formal caregivers was also considered positive. The use of the SOC model contributed to a deeper understanding of technology use in relation to the model’s central components: comprehensibility, manageability, and meaningfulness. This contributes to explain the use of technology-based HC among older persons with CI. Therefore, in the development of mobile health technologies, it is imperative to implement the preferences of older persons with CI and preferably include them as co-designers to improve the health technologies to be used in future home care.

## Figures and Tables

**Figure 1 ijerph-18-11332-f001:**
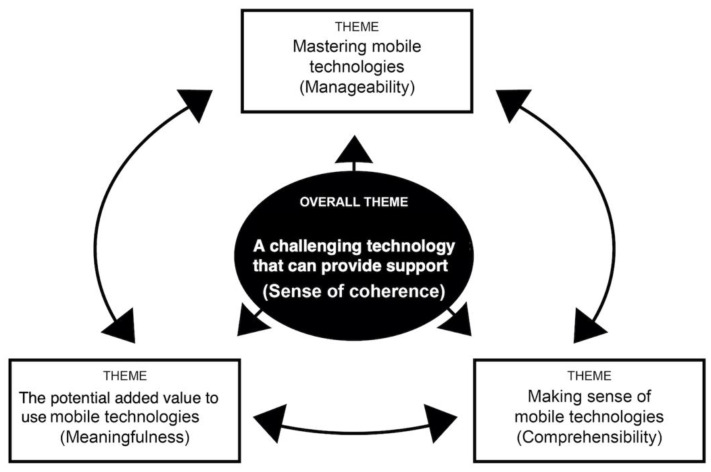
Thematic map of the overall theme and themes corresponding to SOC, including the three central components: comprehensibility, manageability, and meaningfulness.

## Data Availability

Interview data not available due to ethical statement, including secrecy of the participants.

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
