# Peer review of "Using a Mobile Application for Health Communication to Facilitate a Sense of Coherence: Experiences of Older Persons with Cognitive Impairment"

_ijerph, 2021, doi:10.3390/ijerph182111332_

Round 1

Reviewer 1 Report

Thank you for the opportunity to review this paper reporting the use of mobile applications for health communication for older people with mild cognitive impairment. Sixteen people over the age of 55y who had been using the mobile applications on a provided tablet PC for more than a year were enrolled from the larger SMART4MD trial and interviewed using semi-structured interviews and thematic analysis. The paper reports the usefulness and user satisfaction with the intervention based on the “sense of coherence” model and its three core concepts, comprehensibility, manageability, and meaningfulness.

It would assist the reader if the concept of sense of coherence – a person's ability to use existing and potential resources to combat stress and promote health, or, coping capacity to deal with everyday life stresses - was defined or explained early in the introduction.

Participants: the inclusion criteria for this study differ slightly from the inclusion criteria for the SMART4MD trial of which they are a part. What was the rationale for this difference?

Participants: one of the inclusion criteria was that the interviewees had been using the mobile application for more than a year. Based on the quotation in the results, it would appear that at least three participants had stopped using the apps on the tablet and or the tablet, “I did not continue”, “I’m using the computer because it is easier than the tablet”, and “I haven’t been that interested”. Did all of the interviewees meet the inclusion criteria? As they were purposively sampled, was it known in advance that some were no longer using the tablet and the apps?

Discussion, para 1, lines 368-369: “In the findings, the mobile application was considered beneficial and helpful but also challenging.” This presents a positive view of the use of the tablets and the apps, but this is not the sense conveyed in the results. The users found its use challenging, and most of the reported comments were negative, with some suggesting that smartphones, computers and even a pencil and paper were better than the tablet and the apps. The only beneficial results are presented in section 3.1.4.1, and even that is confusing. So was the intervention beneficial? The challenges seem to outweigh the minimal benefits.

This positive interpretation of the findings is repeated in the Conclusions section. The reader is left with the impression that the tablet and the apps did not have the desired or expected impact and that this issue has been largely avoided in the Discussion and Conclusions.  This is a major weakness of the paper.

Did the intervention lead to a sense of cohesion in the subjects? It would appear not.

The paper needs significant English editing to clarify the meaning of some sentences and phrases and improve the readability of the paper. For example, lines 47-49, “Research studying older persons and HC has shown health information preferred to be credible and easily understood.” and lines 301-302, “According to the participants, using other mobile technologies to get access to healthcare easier had perceived benefits, yet they had to wait in telephone queues.”  

Author Response

Reply reviewer 1:

Comments and Suggestions for Authors (replies in red)

Thank you for the opportunity to review this paper reporting the use of mobile applications for health communication for older people with mild cognitive impairment. Sixteen people over the age of 55y who had been using the mobile applications on a provided tablet PC for more than a year were enrolled from the larger SMART4MD trial and interviewed using semi-structured interviews and thematic analysis. The paper reports the usefulness and user satisfaction with the intervention based on the “sense of coherence” model and its three core concepts, comprehensibility, manageability, and meaningfulness.

It would assist the reader if the concept of sense of coherence – a person's ability to use existing and potential resources to combat stress and promote health, or, coping capacity to deal with everyday life stresses - was defined or explained early in the introduction.
Reply: Thank you for this comment. We have now clarified explanation of sense of coherence in the introduction.

Participants: the inclusion criteria for this study differ slightly from the inclusion criteria for the SMART4MD trial of which they are a part. What was the rationale for this difference?
Reply: The study focused on a subgroup of the study (participants from the intervention group), further implying cognitive impairment (MMSE score 20-26) and experience of using the application.   

Participants: one of the inclusion criteria was that the interviewees had been using the mobile application for more than a year. Based on the quotation in the results, it would appear that at least three participants had stopped using the apps on the tablet and or the tablet, “I did not continue”, “I’m using the computer because it is easier than the tablet”, and “I haven’t been that interested”. Did all of the interviewees meet the inclusion criteria? As they were purposively sampled, was it known in advance that some were no longer using the tablet and the apps?
Reply: The participants had had the tablet/intervention for a year’s time, which implies experience of using the application. The inclusion criteria were not active use, which meant that some participants used the application less than others.   

Discussion, para 1, lines 368-369: “In the findings, the mobile application was considered beneficial and helpful but also challenging.” This presents a positive view of the use of the tablets and the apps, but this is not the sense conveyed in the results. The users found its use challenging, and most of the reported comments were negative, with some suggesting that smartphones, computers and even a pencil and paper were better than the tablet and the apps. The only beneficial results are presented in section 3.1.4.1, and even that is confusing. So was the intervention beneficial? The challenges seem to outweigh the minimal benefits.
Reply: Clarified to potential benefits in discussion to emphasize these findings.

This positive interpretation of the findings is repeated in the Conclusions section. The reader is left with the impression that the tablet and the apps did not have the desired or expected impact and that this issue has been largely avoided in the Discussion and Conclusions. This is a major weakness of the paper.
Reply: Clarified to potential benefits in conclusion to emphasize these findings.

Did the intervention lead to a sense of cohesion in the subjects? It would appear not.
Reply:  Thank you for this comment. The study applies the sense of coherence model to probe understanding and use of technology related to comprehensibility, manageability and meaningfulness. Not whether using technology was coherent of the study population (which was mirrored in the results). This is also clarified in the conclusion.

The paper needs significant English editing to clarify the meaning of some sentences and phrases and improve the readability of the paper. For example, lines 47-49, “Research studying older persons and HC has shown health information preferred to be credible and easily understood.” and lines 301-302, “According to the participants, using other mobile technologies to get access to healthcare easier had perceived benefits, yet they had to wait in telephone queues.”
Reply: Thank you for this valuable comment. The manuscript has now gone through an additional English review to improve the language.

Reviewer 2 Report

This manuscript describes an study about how older persons with cognitive impairment experienced the use of a mobile application developed in the SMART4MD project. 16 individuals were interviewed, and their responses were analysed and highlighted under the Sense of Coherence model.

The study is well-written, has a proper structure and posses some interest. Results are strongly aligned with known state of the art and previous work. Thus, the solely major flaw that may be stated is that contributions of this paper are not clear enough. Conclusions and remarks have been already pointed out by others, and in this work authors support those assumptions. Innovative contributions (what this work add to the prior knowledge) should be more clearly described. 

Additionally, the mobile application the interviews deal with should be more deeply explained, in order to understand the extent of its functionality, screens, and technology skills required to use it.

Furthermore, minor comments should be fixed:
1. Acronym HC in the abstract not explained.
2. Line 103: Appendix A > B
3. Line 108: Appendix B > C
4. Line 557: my technology > by technology
4. References should represent a more updated state-of-art. There are many references from 2016 and prior.

Author Response

Comments and Suggestions for Authors (replies in red)

This manuscript describes an study about how older persons with cognitive impairment experienced the use of a mobile application developed in the SMART4MD project. 16 individuals were interviewed, and their responses were analysed and highlighted under the Sense of Coherence model.

The study is well-written, has a proper structure and posses some interest. Results are strongly aligned with known state of the art and previous work. Thus, the solely major flaw that may be stated is that contributions of this paper are not clear enough. Conclusions and remarks have been already pointed out by others, and in this work authors support those assumptions. Innovative contributions (what this work add to the prior knowledge) should be more clearly described.
Reply: Added contributions to previous work of studies related to older persons with cognitive impairment and technology use in conclusions. Also, the clarified the use of the SOC model as a method to understand the use of technology, which is stated in the end of the discussion and in conclusions.

Additionally, the mobile application the interviews deal with should be more deeply explained, in order to understand the extent of its functionality, screens, and technology skills required to use it.
Reply: Revised information about the application in 2 .2, study context. The application is further described in Anderberg et al. (2019).

Furthermore, minor comments should be fixed: 1. Acronym HC in the abstract not explained. 2. Line 103: Appendix A > B
Reply: Edited in manuscript

  1. Line 108: Appendix B > C
    Reply: Edited in manuscript
  2. Line 557: my technology > by technology
    Reply: Edited in manuscript
  3. References should represent a more updated state-of- art. There are many references from 2016 and prior.
    Reply: Thank you for this comment. I have no updated some of the references to more recent in the discussion as well as removed a few older ones.

Round 2

Reviewer 1 Report

Thank you for the opportunity to review the revised version of this paper. Several of the previous queries have not been adequately addressed.

  1. Previous query:

The paper needs significant English editing to clarify the meaning of some sentences and phrases and improve the readability of the paper.

Reply:

Thank you for this valuable comment. The manuscript has now gone through an additional English review to improve the language.

New query:

This doesn’t appear to have been done with care. In the introduction alone, at least eight sentences need careful revision, starting with the first.  

  1. Previous query:

Participants: one of the inclusion criteria was that the interviewees had been using the mobile application for more than a year. Based on the quotation in the results, it would appear that at least three participants had stopped using the apps on the tablet and or the tablet, “I did not continue”, “I’m using the computer because it is easier than the tablet”, and “I haven’t been that interested”. Did all of the interviewees meet the inclusion criteria? As they were purposively sampled, was it known in advance that some were no longer using the tablet and the apps?

Reply:

The participants had had the tablet/intervention for a year’s time, which implies experience of using the application. The inclusion criteria were not active use, which meant that some participants used the application less than others.  

  1. New query:

The wording in the original submission was, “…and been using the application for more than one year”. This is very different to the amended version, “…and access to the mobile application for more than one year prior to the study”. If the correct version is the latter statement, then this adds further support to the following query.

Previous query:

Discussion, para 1, lines 368-369: “In the findings, the mobile application was considered beneficial and helpful but also challenging.” This presents a positive view of the use of the tablets and the apps, but this is not the sense conveyed in the results. The users found its use challenging, and most of the reported comments were negative, with some suggesting that smartphones, computers and even a pencil and paper were better than the tablet and the apps. The only beneficial results are presented in section 3.1.4.1, and even that is confusing. So was the intervention beneficial? The challenges seem to outweigh the minimal benefits.

and

This positive interpretation of the findings is repeated in the Conclusions section. The reader is left with the impression that the tablet and the apps did not have the desired or expected impact and that this issue has been largely avoided in the Discussion and Conclusions. This is a major weakness of the paper.

Reply:

Clarified to potential benefits in discussion to emphasize these findings.

And

Clarified to potential benefits in conclusion to emphasize these findings.

  1. New query:

Respectfully, the addition of the word potential does not address the query. There was very little that was positive in the results as presented. Of 16 interviewees’ responses presented in the Results, only one is positive. The reality is that while the authors perceive it to have potential benefits, the results don’t support this. The interviewees, based on the presented results, found its use challenging. Most of the reported comments were negative, suggesting that smartphones, computers and even a pencil and paper were better than the tablet and the apps, and several stopped using it. This is not the same as saying, “In the findings, the mobile application was considered beneficial and helpful but also challenging.” The interpretation of the results and the conclusions reached should be valid. They currently appear to reflect the authors’ opinion or view point, rather than a scholarly interpretation of the results as presented.

Author Response

Please see below for the reply to the reviewers’ comments. Replies in red.

Thank you for the opportunity to review the revised version of this paper. Several of the previous queries have not been adequately addressed.

  1. Previous query:

The paper needs significant English editing to clarify the meaning of some sentences and phrases and improve the readability of the paper.

Reply:

Thank you for this valuable comment. The manuscript has now gone through an additional English review to improve the language.

New query:

This doesn’t appear to have been done with care. In the introduction alone, at least eight sentences need careful revision, starting with the first.
Reply: Thank you for this comment. We have now revised the English a second time.

  1. Previous query:

Can be Must be Not improved improved applicable

(x) () () ()

(x) () () () (x) () () () (x) () () () () () (x) ()

Yes

https://susy.mdpi.com/user/manuscripts/review/20707625?report=15028437 Sidan 3 av 6

MDPI | Reply review report 2021-10-12 07:57

Participants: one of the inclusion criteria was that the interviewees had been using the mobile application for more than a year. Based on the quotation in the results, it would appear that at least three participants had stopped using the apps on the tablet and or the tablet, “I did not continue”, “I’m using the computer because it is easier than the tablet”, and “I haven’t been that interested”. Did all of the interviewees meet the inclusion criteria? As they were purposively sampled, was it known in advance that some were no longer using the tablet and the apps?

Reply:

The participants had had the tablet/intervention for a year’s time, which implies experience of using the application. The inclusion criteria were not active use, which meant that some participants used the application less than others.

  1. New query:

The wording in the original submission was, “...and been using the application for more than one year”. This is very different to the amended version, “...and access to the mobile application for more than one year prior to the study”. If the correct version is the latter statement, then this adds further support to the following query.

Previous query:

Discussion, para 1, lines 368-369: “In the findings, the mobile application was considered beneficial and helpful but also challenging.” This presents a positive view of the use of the tablets and the apps, but this is not the sense conveyed in the results. The users found its use challenging, and most of the reported comments were negative, with some suggesting that smartphones, computers and even a pencil and paper were better than the tablet and the apps. The only beneficial results are presented in section 3.1.4.1, and even that is confusing. So was the intervention beneficial? The challenges seem to outweigh the minimal benefits.

and

This positive interpretation of the findings is repeated in the Conclusions section. The reader is left with the impression that the tablet and the apps did not have the desired or expected impact and that this issue has been largely avoided in the Discussion and Conclusions. This is a major weakness of the paper.

Reply:

Clarified to potential benefits in discussion to emphasize these findings.

And

Clarified to potential benefits in conclusion to emphasize these findings.

  1. New query:

Respectfully, the addition of the word potential does not address the query. There was very little that was positive in the results as presented. Of 16 interviewees’ responses presented in the Results, only one is positive. The reality is that while the authors perceive it to have potential benefits, the results don’t support this. The interviewees, based on the presented results, found its use challenging. Most of the reported comments were negative, suggesting that smartphones, computers and even a pencil and paper were better than the tablet and the apps, and several stopped using it. This is not the same as saying, “In the findings, the mobile application was considered beneficial and helpful but also challenging.” The interpretation of the results and the conclusions reached should be valid. They currently appear to reflect the authors’ opinion or view point, rather than a scholarly interpretation of the results as presented.
Reply: Thank you for this valuable comment. The overall theme, discussion and conclusion have now been amended to better reflect the result, although both views must be acknowledged. The amended overall theme is suggested to now better reflect the emphasis on negative opinions in the results.
